# Experimental Investigation on the Influence of Target Physical Properties on an Impinging Plasma Jet

**Emanuele Simoncelli** [1] , **Augusto Stancampiano** [1,2] , **Marco Boselli** [1] , **Matteo Gherardi** [1,3] **and Vittorio Colombo** [1,3,*]

1   Department of Industrial Engineering (DIN), Alma Mater Studiorum-Università di Bologna, 40131 Bologna, Italy; emanuele.simoncelli@unibo.it (E.S.); augusto.stancampiano@univ-orleans.fr (A.S.); marco.boselli@unibo.it (M.B.); matteo.gherardi4@unibo.it (M.G.)
2   Presently at GREMI, UMR7344 CNRS/Université d'Orléans, 45067 Orléans, France
3   CIRI—Advanced Applications in Mechanical Engineering & Materials Technology, Alma Mater Studiorum—Università di Bologna, 40131 Bologna, Italy
*   Correspondence: vittorio.colombo@unibo.it; Tel.: +39-051-209-3978

**Abstract:** The present work aims to investigate the interaction between a plasma jet and targets with different physical properties. Electrical, morphological and fluid-dynamic characterizations were performed on a plasma jet impinging on metal, dielectric and liquid substrates by means of Intensified Charge-Coupled Device (ICCD) and high-speed Schlieren imaging techniques. The results highlight how the light emission of the discharge, its time behavior and morphology, and the plasma-induced turbulence in the flow are affected by the nature of the target. Surprisingly, the liquid target induces the formation of turbulent fronts in the gas flow similar to the metal target, although the dissipated power in the former case is lower than in the latter. On the other hand, the propagation velocity of the turbulent front is independent of the target nature and it is affected only by the working gas flow rate.

**Keywords:** impinging jet; metal/dielectric/liquid substrate; ICCD imaging; high-speed Schlieren imaging; plasma discharge morphology; turbulence

## 1. Introduction

The physical and chemical properties of a cold atmospheric pressure plasma (CAP) jet are not uniquely dependent on the plasma source configuration and operational parameters, but also on the target characteristics. The complex mutual interaction between the plasma and the target has recently been the subject of an increasing number of papers, investigating how the targets significantly affect the plasma properties, such as fluid-dynamics [1–3], electromagnetic field [4,5], reactive and excited species production and distribution [6], and ionization front velocity and propagation [1]. Especially, electrical properties, such as conductivity and potential, play a major role [4,7]. During direct plasma treatment, the target becomes part of the transient electrical circuit, connecting the power supply, the plasma source, the plasma, the target and the return to ground [8,9]. This means that the target's electrical parameters (conductivity, potential, etc.) play a major role in determining treatment conditions. Darny et al. revealed how, with conductive substrates like metal, the electric field adjacent to the substrate is enhanced, allowing for a restrike discharge that in turn can greatly enhance the production of reactive species that play an important role in biological applications [10]. Modelling studies by Norberg et al. also predicted similar modifications of the electric field based on substrate conductivity [11]. As reported by Yuanfu et al. [7], the production of OH radicals over a metal substrate can be ten times higher than in the case of a dielectric substrate. In light of the wide range of industrial, biomedical and agricultural applications, the interaction of atmospheric plasma

jets with liquids can be of crucial interest to the scientific community. Liquid substrates, having both capacitive and conductive components, show a hybrid behavior between the purely dielectric and the purely conductive materials [1,7]. This behavior is modulated by the conductivity of liquid solutions that can greatly vary in a range of several order of magnitude ($10^{-7}$–$10^{-2}$ S/cm).

In this frame, the present work provides a direct comparison of the behavior of an atmospheric pressure plasma jet when interacting with a conductive liquid solution, or with dielectric and metal substrates, using ICCD and high-speed Schlieren as imaging techniques. The aim is to gather new insight into the fluid-dynamic behavior of this configuration; an aspect which has not been deeply investigated in the literature, despite being of great influence in plasma assisted processes.

## 2. Materials and Methods

### 2.1. Plasma Source

The plasma source adopted in this work was a single electrode plasma jet developed at the University of Bologna, Italy, and already described, characterized, and applied in previous works by Colombo et al. [1,2,12]. The plasma source was driven by a commercial nanosecond pulse generator (FPG 20-1NMK, FID GmbH). The electric conditions used for all experiments were 15 kV as the peak voltage (PV) and 125Hz as the pulse repetition frequency (PRF). The main voltage pulse supplied by the generator lasted around 35 ns with 10 ns as rise time. A residual damping signal was recorded up to 100 ns, thus the discharge event lasted less than 100 ns. The ignition of the discharge was repeated every 8 ms as a consequence of imposed PRF.

The high-voltage pin electrode (a stainless steel needle; Ø 0.3 mm) was centered inside a dielectric channel, and a flow rate of 3 slpm of helium gas (99.999% pure) was injected through a 12 hole (Ø 0.3 mm) diffuser. The plasma was ejected from the source into the surrounding atmosphere through an orifice with a diameter of 1 mm, producing a visible plasma plume and possibly interacting with a substrate. All experiments were performed in controlled ambient air at 30 °C. To estimate the electrical power dissipated by the plasma source, voltage and current waveforms were recorded on the high-voltage connection powering the plasma source by means of a high-voltage probe (Tektronix P6015A, sensitivity of 0.018%) and a current probe (Pearson 6585, sensitivity of 1%) connected to an oscilloscope (Tektronix DPO40034). The power density was evaluated using the following formula:

$$SPD = PRF \cdot \int V \cdot I dt \tag{1}$$

where *PRF* is the pulse repetition frequency, *V* is the applied voltage, and *I* is the current (the voltage and current signal are related to the main positive peak).

### 2.2. Substrates

The plasma source was positioned vertically at 10 mm (fixed gap) above the substrate surface. Different targets characterized by different conductivities were selected for this study, ranging from metal and liquid to dielectric substrates. A stainless steel plate (7.6 cm × 7.6 cm × 1 cm) was chosen as the target with almost infinite conductivity (~$3.7 \times 10^{11}$ μS/cm). On the other hand, to simulate a non-conductive substrate, a 7.6 cm × 7.6 cm × 1 cm PVC plate was used as the dielectric substrate (~$10^{-17}$ μS/cm). As far as the liquid substrate was concerned, since the electrical conductivity of the target affects the plasma characteristics [13,14], and in turn the plasma treatment may alter the electrical conductivity of the treated liquid solutions [15,16], the liquid target was prepared as a phosphate buffer solution (made by dissolving sodium phosphate dibasic ($Na_2HPO_4$) and potassium phosphate monobasic ($KH_2PO_4$) in distilled water). A solution characterized by an electrical conductivity of 119 μS/cm with a pH of 7.2 was realized. The physical properties (conductivity and pH) were monitored and remained unaltered during the experiments. The solution volume was 120 mL contained in a vessel (7.6 cm × 7.6 cm × 2 cm) with quartz sidewalls and an aluminum bottom. Since the metal

substrate and the aluminum bottom of the liquid substrate vessel were connected to ground through a low impedance electrical connection, both could be considered at ground potential. The dielectric substrate was instead positioned on a grounded metal plate to control and fix the associated capacitance. According to the literature [9,17], the equivalent electrical circuits associated with the three substrates result in a single resistance for the metal substrate, a single capacitance for the dielectric substrate, and a resistance and capacitance in parallel for the liquid substrate. Presenting very different electrical characteristics, the three substrates are a good representation of a wide range of possible targets.

## 2.3. Diagnostic Techniques

The morphology and the time evolution of the plasma discharge interacting with each substrate were investigated by means of an ICCD camera (Princeton Instruments PIMAX3, spectral response 180–900 nm) equipped with a conventional macro lens (Sigma Dg-Ex-APO-If 180 Mm/F3.5, spectral response 380–900 nm). To synchronize the image acquisition with the discharge event, the synchronization of the camera gating, driven by the voltage pulse, was performed by employing a delay generator (BNC 575 digital pulse/delay generator) and taking into account all possible signal transmission delays, as already described in the literature [12]. An overview of the iCCD imaging setup is showed in Figure 1.

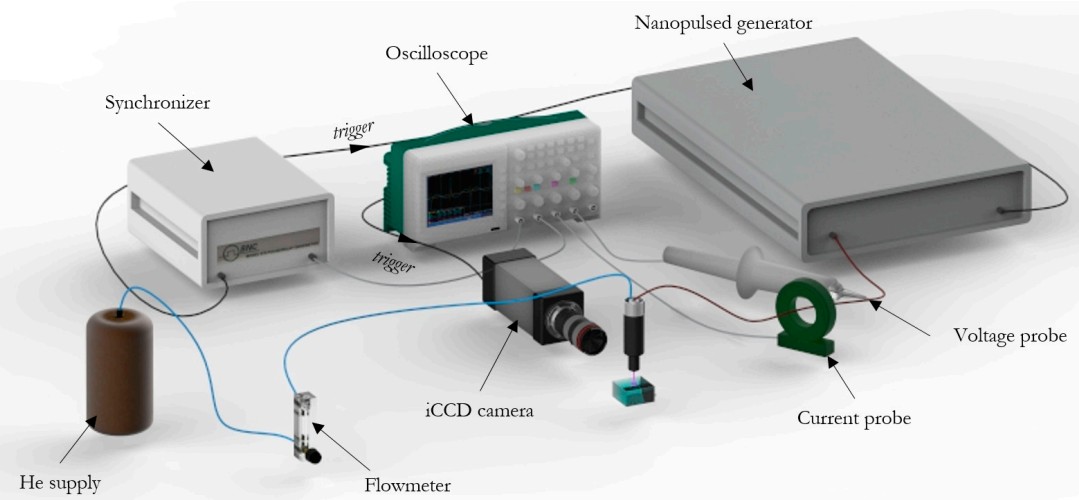

**Figure 1.** ICCD imaging setup.

The overall emission intensity, produced by the plasma discharge during the main voltage pulse, was acquired by imposing both the camera gate exposure (35 ns) and the duration of the voltage pulse. The images were accumulated 30 times with a gain factor set at 50. The time evolution of the plasma discharge during the whole voltage pulse was investigated through the capture of sequential 10 ns camera gate exposures (each frame resulting from 30 accumulations). The first ICCD gate opening (0 ns) was imposed, so as to center on the start of the rising front of the voltage pulse (see Figure 3, top). The ICCD camera gates were superimposed on the excitation voltage waveforms. The reproducibility of the discharge was first verified with single shot (no accumulations) acquisitions (data not shown).

The characterization of the fluid-dynamic of the impinging jet was performed using the Schlieren imaging technique [18]. The Schlieren setup is shown in Figure 2 and was composed of a 450 W ozone free xenon lamp (Newport-Oriel 66355 Simplicity Arc Source) as a light source, a slit and an iris diaphragm, two parabolic mirrors with a focal length of 1 m, a knife edge positioned vertically because the highest gradients of the refractive index around the axis of the jet were horizontal, and a high-speed camera (Memrecam GX-3 NAC image technology). The camera was operated at 8000 fps with 1/200,000 s shutter time. The plasma source was positioned halfway between the two parabolic mirrors.

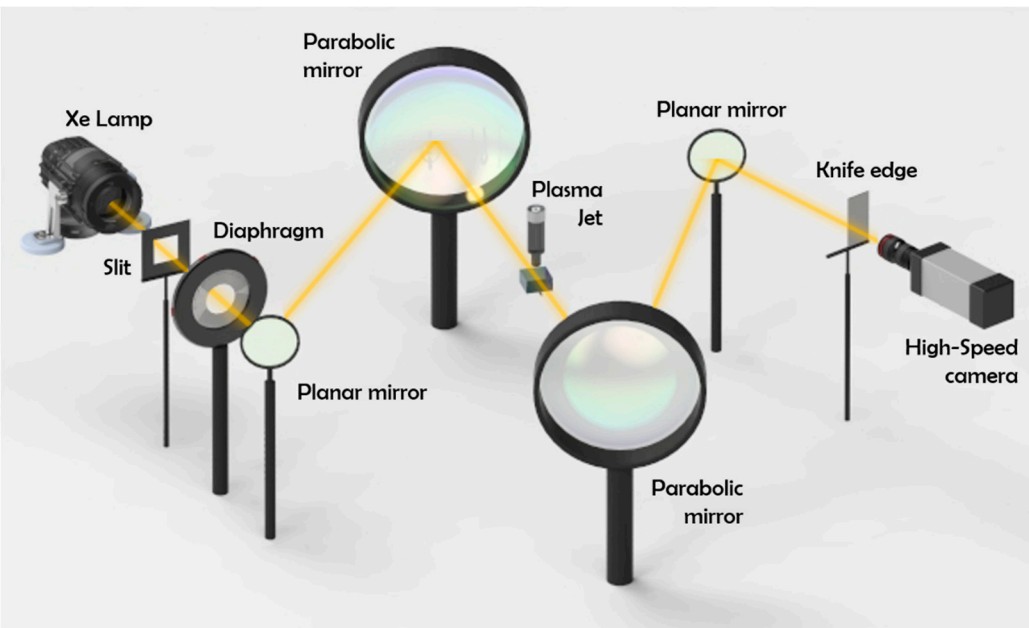

**Figure 2.** Schlieren imaging setup.

## 3. Results

### 3.1. Electrical and Time-Resolved ICCD Characterization of the Plasma Jet Impinging on Different Substrates

Table 1 shows the measured values of the electrical power, dissipated by the plasma source interacting with metal, dielectric and liquid substrates, respectively. Although the input operating conditions were the same for all investigated cases, the plasma jet impinging on a metal target dissipated the highest power (0.434 W) in comparison to the other targets.

**Table 1.** Electrical power dissipated by the plasma source for metal, dielectric and liquid substrates.

|  | Metal | Dielectric | Liquid (119 µS/cm) |
|---|---|---|---|
| Power [W] | 0.434 | 0.313 | 0.255 |

In this section, the results for the ICCD imaging of the plasma jet plume impinging on different substrates are presented.

Figure 3 shows, for each condition, a single accumulated image of the discharge obtained by capturing the light intensity emitted during the whole main voltage pulse (duration of ~35 ns). The acquisitions highlight how the plasma, and therefore its Visible–Near InfraRed (Vis-NIR) light emission, was highly influenced by the nature of the substrate. The strongest intensity and largest width of the plasma columns were recorded for the case of a metal target (fourth image from the left in Figure 3). On the other hand, the lowest intensity was observed when the plasma jet impinged on the liquid substrate (second image from the left in Figure 3). Furthermore, a spreading of the plasma across the target surface, known as surface ionization wave (SIW), was clearly observed only in the case of the dielectric substrate. For the liquid substrate, the plasma discharge appeared to be focused on a small area, corresponding to the dimple created in the liquid by the jet effluent (first image on the left in Figure 3).

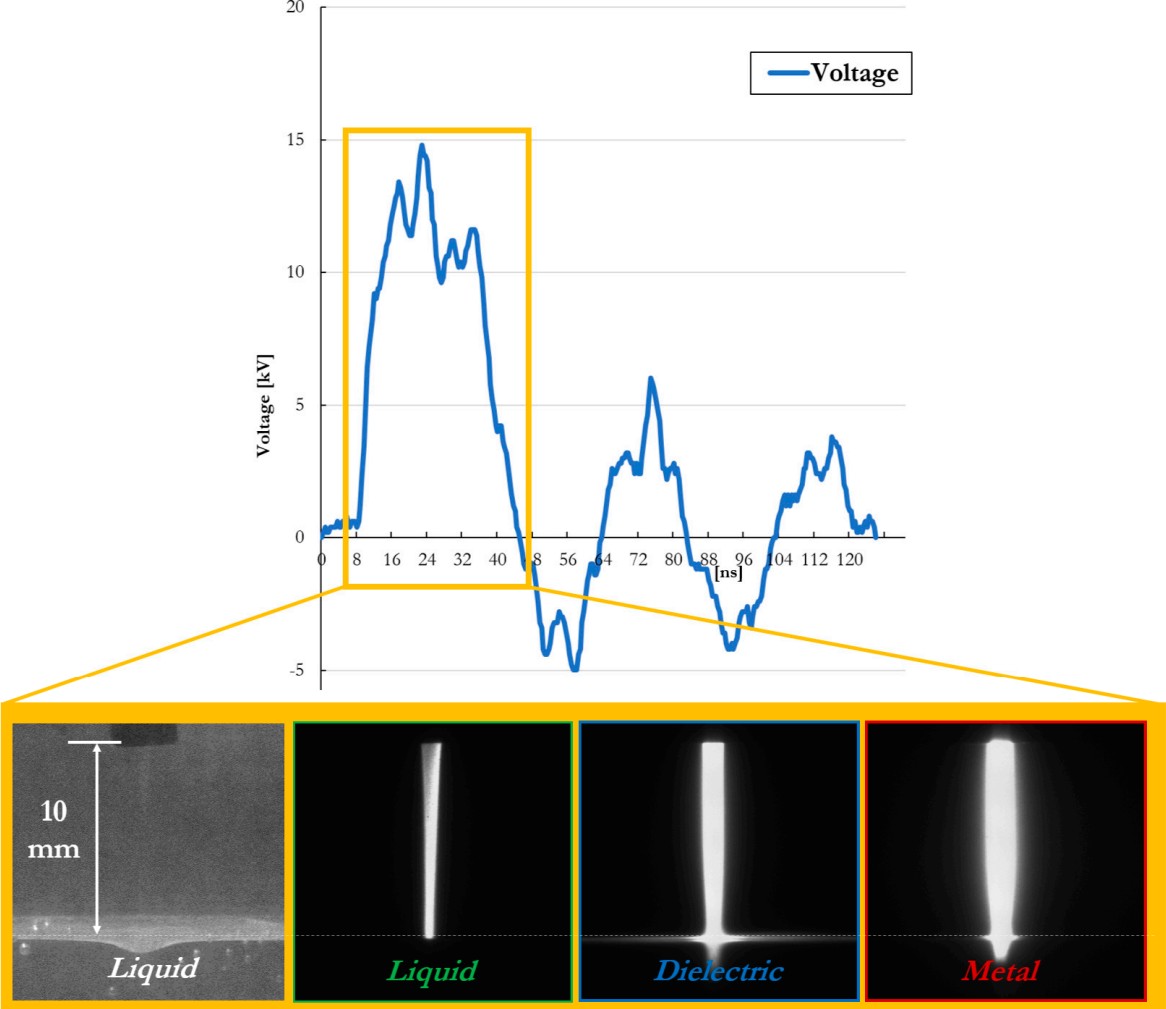

**Figure 3.** ICCD images of the discharge emission related to the dielectric, liquid and metal substrates. (The first image on the left, taken with a longer exposure time, is presented to show the dimple formation on the liquid substrate).

In Figure 4, the sequences of ICCD images (10 ns exposure) for both dielectric and metal cases are shown.

The acquisitions (Figure 4) highlight how the temporal evolution of the plasma Vis-NIR light emission was influenced by the nature of the target. The propagation velocity of the ionization front was estimated to be $\geq 2 \times 10^8$ cm/s, as the crossing of the gap took place in the first two acquisitions.

For both substrates, the peak of the emission intensity was achieved in the acquisition at 20 ns, corresponding to the reach of the maximum applied voltage (Figure 4, third acquisition from the left for both cases). In the case of the dielectric target, the images show how the Vis-NIR light emission remained approximately constant during the whole voltage pulse. In contrast, on the metal substrate, the plasma emission significantly changed with time; therefore, the higher the applied voltage, the more intense the light emission.

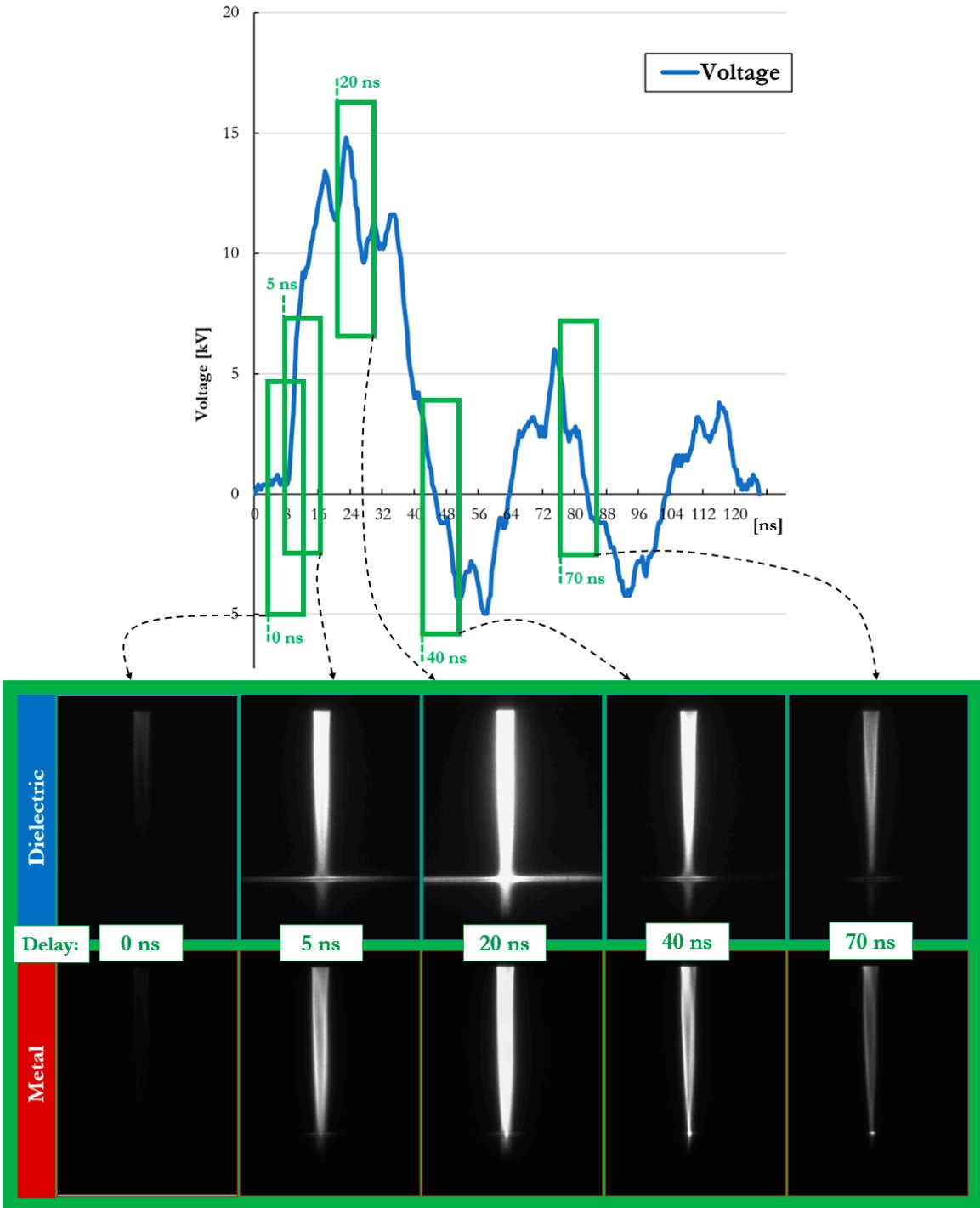

**Figure 4.** Time-resolved ICCD imaging of the plasma jet plume impinging on the dielectric and metal substrates.

The ICCD images in Figure 4 reveal how, in the case of a metal substrate, the plasma plume did not show a clear SIW formation, but the discharge was quite focused on a point on the surface. On the other hand, in Figure 3 the image related to the metal substrate shows a weak SIW upon the surface, as a consequence of a time integration of 35 ns corresponding to the imposed exposure time. As shown by the frames 5 and 20 ns in Figure 4, (metal substrate) photons emission covered a wider area compared to the frames of 40 and 70 ns.

In the ICCD time-resolved images shown in Figure 4, the SIW formation and development were clearly enhanced in the case of the plasma jet interacting with the dielectric target, due to its higher capacitance. The dielectric surface was charged more than the metal one, favoring a higher spreading of the plume over it.

### 3.2. Time-Resolved Schlieren Characterization of the Plasma Jet Impinging on Different Substrates

Figure 5 presents results for the high-speed Schlieren imaging of the plasma jet plume impinging on the dielectric, liquid and metal substrates. The frames were selected with the aim of emphasizing the most important steps of spatial evolution of the turbulent front induced by the discharge, as follows: the discharge event, the turbulent front exiting the nozzle, the turbulent front approaching the substrate, its impact with the substrate surface, and its expansion upon the surface.

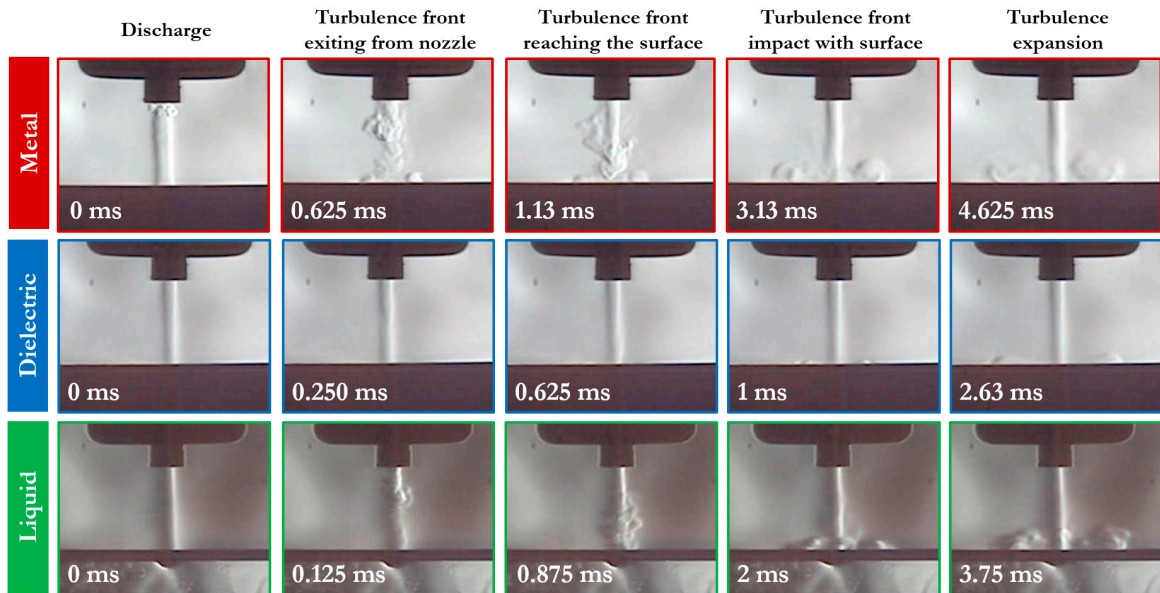

**Figure 5.** Time-resolved Schlieren images of the plasma jet plume impinging on the metal, dielectric and liquid substrates. The reported time values are indicative of the time lapse from the ns-discharge to the start of the corresponding acquisition of the high-speed camera.

The duration of the plasma discharge emission (around 100 ns, as described in Section 2.1) was entirely captured in the first frame (0 ms), since the exposure of the high-speed camera was set at 0.005 ms. The time values reported on each frame were indicative of the time lapse between the acquisition of the first frame and those that followed.

In the case of the metal target, a turbulent front was observed at the outlet of the plasma jet in the first image, while for all the other cases, in the first frame (0 ms) the He gas flow seemed completely laminar, similar to the case of He gas flow without plasma ignition (data not presented). For dielectric and liquid substrates, the acquisitions at 0 ms showed how during the plasma discharge, the He gas flow was laminar with no flow modifications visible. Nonetheless, a significant perturbation of the He gas flow was clearly visible in the following frames, several tens of microseconds after the plasma discharge and the imposed voltage pulse ended. The turbulent phenomena is thus ignited by the discharge event, and its dynamic behavior is directly affected by the nature of the target.

At 0.25 ms after the end of the plasma discharge, in all investigated cases, a transient turbulent structure appeared as a consequence of an induced alteration of the helium gas characteristics, such as temperature and density, inside the plasma source during the discharge. The turbulent front coming out of the nozzle propagated downstream with a velocity close to that of the gas flow (~60 m/s). Once this transient turbulent structure reached the target surface, eddies were generated and propagated above the surface, away from the column axis. In all investigated cases, a laminar flow was finally

re-established (data not shown) in the He column before the next discharge was ignited (8 ms period for PRF 125 Hz).

The strongest turbulent front was observed in the case of the metal substrate. The turbulent structures remained visible more than 5 ms after the discharge event (data not shown). In the case of the liquid substrate, the turbulent phenomenon that was induced was well recognizable. On the other hand, when the plasma jet impinged on a dielectric substrate, the induced turbulence was weaker, and the turbulent structures propagated over the surface were rapidly dissipated by interaction with the surrounding air. While the amplitude and the intensity of the turbulence were dependent on the target nature, the velocity of propagation of the turbulent front appeared independent of the target nature and governed by the gas flow rate only.

Finally, the gas impinging on the liquid substrate caused the formation of an axisymmetric dimple on its surface. The effects of impingement of a gas jet on a liquid surface have been studied in detail in different scientific and industrial fields [19–21]. As described in previous work [1], the impact of the turbulent front upon the liquid surface causes a variation of the dimension of the dimple in time.

## 4. Discussion

According to the results, the influence of the electrical properties of the material on the discharge characteristics of an impinging plasma jet is clearly visible. Materials with very low conductivity, such as the dielectric PVC plate (~$10^{-17}$ μS/cm), favor the accumulation of charges on their surface. The total charge accumulated increases with the capacitance of the target [22] and, in our case, was significantly higher for the dielectric case in comparison with the metal and liquid substrates. The charge accumulation in turn induces the development of a radial electric field driving the formation of a SIW [11]. This results in the symmetrical radial expansion over the surface of the discharge, as clearly visible in Figure 4. SIW phenomenon is undetectable in the metal and the liquid cases, due to their much higher conductivity (~$3.7 \times 10^{11}$ μS/cm for the metal and 119 μS/cm for the liquid) that prevents the accumulation of charges on the surface.

On the other hand, the conductivity of the substrate also limits the power deposited by the plasma discharge. After the impinging of the ionization front on the substrate surface, a highly conductive channel connects the high-voltage electrode inside the jet source to the grounded substrate [10]. Once this connection occurs, the current passing through the plasma channel is limited by the conductivity of the target and the capacitance associated with it. As demonstrated by the power measurements (Table 1), the metal substrate, with a conductivity several order of magnitude higher than the other samples, recorded the highest dissipated power, albeit the values of measured power were in the same order of magnitude for all investigated cases. Moreover, although the liquid, with a higher conductivity, presented the lowest power, we hypothesize that other phenomena and factors, such as the target capacitance (higher for the dielectric), could have played a key role in the frame of ns-discharge interacting with a target. As was visible in the dielectric case, the plasma discharge propagated over the target surface and the emission intensity increased to t = 20 ns when the voltage pulse reached its peak. This suggests that the dielectric substrate, due to its capacitance, was still charging and, therefore, only partially limiting the current flow in the conductive channel. The liquid substrate on the other hand, characterized by a conductivity much lower than the metal (nearly 11 orders of magnitude) and a capacitance much lower than the dielectric, presented a singular behavior; no SIW was formed and there was a limitation of the deposited power. It must be noted that the liquid substrate is also the only substrate among those investigated to induce a partial modification of the surrounding gas composition due to evaporation. As reported by Ji et al. [23] in a similar configuration, the presence of a liquid film on a target presents a reduced impact on the characteristics of the plasma discharge propagation and intensity, while on the other hand, it can greatly enhance the production of OH radicals in close proximity to the target surface.

Concerning the plasma-induced alteration of the gas flow, the turbulent front could be clearly seen in all considered cases, but with different characteristics. In the case of the metal substrate, a turbulent

alteration of the gas flow was visible from the discharge event (frame 0 ms in Figure 5). Later, a turbulent front propagated downstream with a velocity comparable with the gas flow's, as already observed in similar conditions in previous work [1]. Since the total expansion of the turbulent front was sufficiently fast enough to observe a re-established laminar He flow before the ignition of the next discharge event, we can confirm that the turbulent front phenomenon is directly associated with a combination of plasma-induced pressure waves and local electrode heating [1,24,25]. Moreover, because the Schlieren images (Figure 5) clearly showed that the magnitude of the turbulent structures varied between the investigated cases, we can hypothesize that turbulent front formation and its propagation is affected by the target characteristics. It may be speculated that there is a relation between turbulences and the target modification of the electric field distribution between the high-voltage electrode inside the plasma source and the targets [23]. In virtue of their conductivities and connections to ground, the metal and liquid targets greatly affected the electric field distribution, while the dielectric PVC target had a relative permittivity comparable to that of air ($\varepsilon_{r-PVC} = 3$), thus having a limited impact.

## 5. Conclusion

The present study highlights, through an ICCD and Schlieren imaging analysis, how the plasma plume changes its morphology and its light intensity as a consequence of the physical properties of the target, and how the fluid-dynamic of the plasma-induced turbulent front is affected by the substrate's nature. The proposed comparison between metal, dielectric and water substrates highlights significant differences between the associated plasma discharges. The electrical characteristics of the substrates may influence not only the plasma discharge propagation and ionization degree, but also the gas flow dynamics. Nanosecond pulsed plasma, interacting with a liquid substrate, presents a singular behavior that is not easily located midway between highly conductive and dielectric materials. Thus, future analysis should be directed to the investigation of substrate materials covering a wider range of conductivities, especially in the range of semiconductive materials. This would certainly help to better identify possible trends and provide better explanations. Concerning the transient turbulence generated in the gas flow by the plasma discharge, it has highlighted a dependence on the target nature. This aspect should be taken into account during plasma treatments, since it may affect through mixing of the gas composition in the plasma region, and may induce pressure fluctuations on the target surface. These aspects may hinder the uniformity of treatments, especially on complex and non-homogeneous surfaces like biological tissues.

The results of this work aim to be considered as one step closer to a full understanding of the complex interaction of non-equilibrium plasma with a substrate.

**Author Contributions:** Conceptualization, E.S., A.S., M.G. and V.C.; Data curation, E.S. and A.S.; Funding acquisition, V.C.; Investigation, E.S., A.S. and M.B.; Methodology, E.S., A.S. and M.B.; Project administration, V.C.; Resources, V.C.; Supervision, M.G. and V.C.; Writing—original draft, E.S., A.S. and V.C.; Writing—review & editing, E.S., A.S., M.G. and V.C.

**Funding:** This research received no external funding.

**Acknowledgments:** The authors would like to acknowledge Romolo Laurita for his contribution in liquid solution preparation and Eng. Alina Bisag for 3D rendering of the experimental setups.

**Conflicts of Interest:** The authors declare no conflicts of interest.

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
