# Peer review of "Experimental Investigation on the Influence of Target Physical Properties on an Impinging Plasma Jet"

_plasma, doi:10.3390/plasma2030029_

Round 1

Reviewer 1 Report

The authors investigate the electrical, optical and flow characteristics of a ns-pulsed atmospheric pressure plasma jet in Helium impinging on substrates with different electrical properties – conductive, dielectric and liquid. By means of high-speed imaging, the authors present time-resolved emission from the plasma and describe the development of a turbulent flow front succeeding the plasma ignition event. Overall, the research presented in this work contributes significantly to the understanding of how plasma-substrate interactions impact the discharge properties. I recommend this work to be accepted for publication after some minor revisions. A list of some points which I believe can improve the presentation of the paper follows:

1.     The paper can benefit from an English language revision and spelling check, below are some instances that require revision

a.     Line 55:  I believe the authors mean ‘damping’ instead of dumping

b.     Line 56:  ‘lasts’ is more appropriate then ‘is lasting’

c.     Line 61:  ‘a substrates’ should be replaced by the singular ‘a substrate’

d.     Line 68: ‘conductivity’ is a more appropriate word choice than ‘conductibility’

e.     Line 211: ‘in turn’ instead of ‘in turns’

f.      Line 242: ‘hypothesize’ instead of ‘hypotyize’

Beyond these examples that stood out, there are several other minor issues in the text. The paper would benefit from a careful review.

2.     The authors mention on line 62 a ‘power density’ measurement although absolute deposited power is reported in the results section. Moreover, the diagram in Figure 1 depicts the current measurement at the input line to the electrode as opposed to the current leaving to the ground. These two values can be appreciably different in some cases. I believe it would be helpful to include some discussion of how the power measurements were obtained, perhaps including some examples of the current waveforms. This may allow clarifying the source of the different power measurements taken with different substrates.  

3.     Authors attribute the differences in power measurements to purely electrical properties of the substrates, however, with the liquid substrate, the presence of comparatively high amounts of water vapor can significantly affect the discharge characteristics, specifically by imparting a quenching effect. This may partially account for the lowest power observed over liquid, although the electrical properties of the substrate is between that of metal and dielectric.

4.     In Figure 4 the authors present time-resolved optical emission recorded on glass and metal substrates. Is a similar measurement available for liquid? If so, it would perhaps be helpful to include these results for completion.

5.     I found the authors’ discussion of surface ionization waves, particularly as it pertains to the metal substrate somewhat confusing. on line 131-132 authors claim a SIW is observed over the metal substrate to a lesser extent, however, on lines 154-155 the authors point out that time-resolved iCCD images do not indicate the presence of SIW on the metal substrate. Some further clarification of this discrepancy would be helpful. Theoretically, a perfectly conductive substrate should not accumulate charge and therefore should not exhibit SIWs. Note that, the presence of optical emission only indicates the presence of excited molecules de-exciting over the region. This does not necessarily prove the presence of an ionization wave. Other factors such as reflection from the substrate (which is clearly present in the images in Figure 3) can also contribute to the recorded image of the discharge.

6.     The authors conclude in lines 243-244 that the intensity of the turbulent front is related to the deposited power. However, the images in Figure 5 indicate a higher rate of turbulence over the liquid substrate (lower power) compared to the dielectric substrate (higher power). This contradiction requires some clarification. For example, the spatial distribution of characteristics such as differences in the magnitudes of local electric fields can play a role in the development of turbulent fronts.

Author Response

WE WOULD LIKE TO THANK THE REFEREES FOR THEIR SUGGESTIONS, AND WE HOPE WE HAVE FULFILLED ALL THEIR REQUESTS. ANSWERS TO THEIR COMMENTS ARE REPORTED BELOW IN BLUE AND THE CHANGES IN THE MANUSCRIPT HAVE BEEN WRITTEN IN RED

The authors investigate the electrical, optical and flow characteristics of a ns-pulsed atmospheric pressure plasma jet in Helium impinging on substrates with different electrical properties – conductive, dielectric and liquid. By means of high-speed imaging, the authors present time-resolved emission from the plasma and describe the development of a turbulent flow front succeeding the plasma ignition event. Overall, the research presented in this work contributes significantly to the understanding of how plasma-substrate interactions impact the discharge properties. I recommend this work to be accepted for publication after some minor revisions. A list of some points which I believe can improve the presentation of the paper follows:

1. The paper can benefit from an English language revision and spelling check, below are some instances that require revision
a. Line 55: I believe the authors mean ‘damping’ instead of dumping
b. Line 56: ‘lasts’ is more appropriate then ‘is lasting’
c. Line 61: ‘a substrates’ should be replaced by the singular ‘a substrate’
d. Line 68: ‘conductivity’ is a more appropriate word choice than ‘conductibility’
e. Line 211: ‘in turn’ instead of ‘in turns’
f. Line 242: ‘hypothesize’ instead of ‘hypotyize’
Beyond these examples that stood out, there are several other minor issues in the text. The paper would benefit from a careful review.
WE WOULD LIKE TO THANK THE REFEREE FOR THE SUGGESTED CORRECTIONS IN THE TEXT. WE MODIFIED AND CONTROLLED THE WHOLE MANUSCRIPT.

2. The authors mention on line 62 a ‘power density’ measurement although absolute deposited power is reported in the results section. Moreover, the diagram in Figure 1 depicts the current measurement at the input line to the electrode as opposed to the current leaving to the ground. These two values can be appreciably different in some cases. I believe it would be helpful to include some discussion of how the power measurements were obtained, perhaps including some examples of the current waveforms. This may allow clarifying the source of the different power measurements taken with different substrates.
FOLLOWING THE SUGGESTIONS OF THE REVIEWER, WE MODIFIED THE PARAGRAPH 2.1 GIVING FURTHER AND MORE DETAILED INFORMATION REGARDING THE MEASUREMENT OF THE POWER DENSITY. WE CONSIDERED TO IMPLEMENT IN THE MANUSCRIPT THE CURRENT WAVEFORMS, BUT AS SHOWN IN THE FOLLOWING GRAPH, THE PROFILE OF CURRENT MAKES HARD TO OBTAIN DIRECT CONSIDERATIONS ON POWER DENSITY, DUE TO THE ELECTRICAL NATURE OF nsDISCHARGES AND WE PREFER TO NOT INCLUDE THEM.

3. Authors attribute the differences in power measurements to purely electrical properties of the substrates, however, with the liquid substrate, the presence of comparatively high amounts of water vapor can significantly affect the discharge characteristics, specifically by imparting a quenching effect. This may partially account for the lowest power observed over liquid, although the electrical properties of the substrate is between that of metal and dielectric.
WE WOULD THANK THE REFEREE TO HAVE UNDERLINED THIS POINT. WE HAVE SUPPERTED BETTER OUR POINT MODIFING THE TEXT (DISCUSSION PARAGRAPH) AND INTRODUCING A REFERENCE WHICH DEMONSTRATES HOW THE PRECENCE OF A LIQUID LAYER AFFECTS THE REACTIVE SPECIES PRODUCTION, SUCH AS OH, MORE THAN THE POWER DENSITY DEPOSITED IN THE PLASMA DISCHARGE.

4. In Figure 4 the authors present time-resolved optical emission recorded on glass and metal substrates. Is a similar measurement available for liquid? If so, it would perhaps be helpful to include these results for completion.
EFFECTIVELY WE DECIDED TO NOT INTRODUCE THESE IMAGES BECAUSE WE DISCUSSED THEM IN AN OUR RECENT PAPER (Stancampiano A, Simoncelli E, Boselli M, Colombo V, Gherardi M. Experimental investigation on the interaction of a nanopulsed plasma jet with a liquid target. Plasma Sources Sci Technol 2018;27:125002(14)). WE CAN UNDERLINE BETTER IN THE INTRODUCTION THE LINK BETWEEN THE PRESENT WORK AND THE MENTIONED PAPER.

5. I found the authors’ discussion of surface ionization waves, particularly as it pertains to the metal substrate somewhat confusing. on line 131-132 authors claim a SIW is observed over the metal substrate to a lesser extent, however, on lines 154-155 the authors point out that time-resolved iCCD images do not indicate the presence of SIW on the metal substrate. Some further clarification of this discrepancy would be helpful. Theoretically, a perfectly conductive substrate should not accumulate charge and therefore should not exhibit SIWs. Note that, the presence of optical emission only indicates the presence of excited molecules de-exciting over the region. This does not necessarily prove the presence of an ionization wave. Other factors such as reflection from the substrate (which is clearly present in the images in Figure 3) can also contribute to the recorded image of the discharge.
WE UNDERSTAND THE REFEREE’S POINT OF VIEW. AS A MATTER OF FACT, WE PREFEREED TO DELETE THE CLAIM AT 131-132 LINE.

6. The authors conclude in lines 243-244 that the intensity of the turbulent front is related to the deposited power. However, the images in Figure 5 indicate a higher rate of turbulence over the liquid substrate (lower power) compared to the dielectric substrate (higher power). This contradiction requires some clarification. For example, the spatial distribution of characteristics such as differences in the magnitudes of local electric fields can play a role in the development of turbulent fronts.
FOLLOWING THE USEFUL SUGGESTIONS OF THE REFEREE, WE MODIFIED THIS PART OF THE DISCUSSION EMPHASIZING THE DIFFERENCES BETWEEN BOTH METAL AND LIQUID SUBSTRATES WITH THE DIELECTRIC ONE.

Reviewer 2 Report

The authors used optical emission spectroscopy and Schlieren imaging to study the effect of target conductivity on an atmospheric pressure plasma jet (APPJ) in helium.

The work is timely and the choice of substrates to cover the extremely wide conductivity window is reasonable. However the authors need to make a major revision before this work may be recommended for publication in PLASMA.

There are countless published works (both experimental and theoretical) that examine (as a major or a minor topic) the effect of target physical properties on the plasma discharge characteristics in APPJ. Many of these works are listed in the references section of the manuscript. The authors have to clarify the important new findings of their work against the background of known facts.  

The authors should provide explanation (even tentative) of their observations, for example, why does the diameter of the visible plume increase from liquid to dielectric to metal target in Fig. 3? Or why, in the case of dielectric, the emission remains almost unchanged even though the voltage is changing in Fig. 4?

In line 195 the authors state that “Although the intensity of the turbulence is affected by the nature of the target, the temporal evolution and dynamic of the front propagation resulted similar for all the investigated cases.” This seems counterintuitive. The discharge characteristics depend on the target conductivity and turbulence is induced by the discharge. Therefore, the evolution of the flow dynamics of the jet should vary depending on the target properties. Please explain.

The authors use the abbreviation “iCCD” throughout the paper except for line 246 where they use “ICCD”. I recommend they use the more common abbreviation “ICCD” everywhere.

There are many misspelled words in the manuscript. For example “corrisponding” in lines 133 and 149, “esteemed” in line 147 (instead of estimated), evinsible” in line 207, “hypotyize” in line 242 etc.

Author Response

WE WOULD LIKE TO THANK THE REFEREES FOR THEIR SUGGESTIONS, AND WE HOPE WE HAVE FULFILLED ALL THEIR REQUESTS. ANSWERS TO THEIR COMMENTS ARE REPORTED BELOW IN BLUE AND THE CHANGES IN THE MANUSCRIPT HAVE BEEN WRITTEN IN RED.

The authors used optical emission spectroscopy and Schlieren imaging to study the effect of target conductivity on an atmospheric pressure plasma jet (APPJ) in helium.
The work is timely and the choice of substrates to cover the extremely wide conductivity window is reasonable. However the authors need to make a major revision before this work may be recommended for publication in PLASMA.
WE WOULD LIKE TO THANK THE REFEREE FOR THE PROPOSED SUGGESTIONS. BELOW, WE ANSWER POINT BY POINT TO THE COMMENTS.

1. There are countless published works (both experimental and theoretical) that examine (as a major or a minor topic) the effect of target physical properties on the plasma discharge characteristics in APPJ. Many of these works are listed in the references section of the manuscript. The authors have to clarify the important new findings of their work against the background of known facts.
AS SUGGESTED BY THE REFEREE, WE MODIFIED THE INTRODUCTION HIGHLIGHTING THE IMPORTANCE AND NOVELTY OF THE PRESENT WORK AND ADDING REFERENCES.

2. The authors should provide explanation (even tentative) of their observations, for example, why does the diameter of the visible plume increase from liquid to dielectric to metal target in Fig. 3? Or why, in the case of dielectric, the emission remains almost unchanged even though the voltage is changing in Fig. 4?
THE DIELECTRIC TARGET HAS A CAPACITIVE ELECTRICAL CHARACTERISTIC THAT FAVOURS CHARGE ACCUMULATION ON ITS SURFACE AND THE WIDENING OF THE ATTACHMENT OF THE PLUME ON THE SURFACE AS WELL AS THE PROPAGATION OF A SIW. THE METAL INSTEAD ACTS LIKE A LOW RESISTANCE WHICH ALLOWS A DIRECT INCREASE IN CURRENT WITH THE APPLIED VOLTAGE. THE INCREASE IN CURRENT LEADS TO A BROADENING OF THE DISCHARGE SECTION. WE CAN SPECULATE THAT THE LIQUID, PRESENTING BOTH CAPACITIVE AND RESISTIVE CHARACTERISTICS, ON ONE SIDE LIMITS THE CHARGE ACCUMULATION ON ITS SURFACE AND THEREFORE THE SPREADING ON THE ATTACHMENT, AND, ON THE OTHER HAND, PRESENTING A
MUCH HIGHER IMPEDANCE COMPARED TO THE METAL, IT LIMITS THE CURRENT FLOW THROUGH THE PLUME.
WE MODIFIED THE DISCUSSION, INSERTING THESE CONSIDERATIONS ON THE DIFFERENCES IN THE MORPHOLOGY OF THE PLASMA PLUME AS A CONSEQUENCE OF THE IMPINGED SUBSTRATE.

3. In line 195 the authors state that “Although the intensity of the turbulence is affected by the nature of the target, the temporal evolution and dynamic of the front propagation resulted similar for all the investigated cases.” This seems counterintuitive. The discharge characteristics depend on the target conductivity and turbulence is induced by the discharge. Therefore, the evolution of the flow dynamics of the jet should vary depending on the target properties. Please explain.
TO EXPLAIN BETTER THE POINT METIONED BY THE REFEREE, WE MODIFIED THE PARAGRAPH 3.2 INTRODUCING THIS SENTENCE “TO RESUME, WHILE THE APLITUDE AND THE INTENSITY OF THE TURBULENCE ARE DEPENDENT ON THE TARGET NATURE, THE VELOCITY OF PROPAGATION OF THE TURBULENT FRONT APPEARS INDIPENDENT OF THE TARGET NATURE AND GOVERNED BY THE GAS FLOWRATE.”

4. The authors use the abbreviation “iCCD” throughout the paper except for line 246 where they use “ICCD”. I recommend they use the more common abbreviation “ICCD” everywhere.
There are many misspelled words in the manuscript. For example “corrisponding” in lines 133 and 149, “esteemed” in line 147 (instead of estimated), evinsible” in line 207, “hypotyize” in line 242 etc.
WE WOULD LIKE TO THANK THE REFEREE FOR THE SUGGESTED CORRECTIONS IN THE TEXT. WE MODIFIED AND CONTROLLED THE WHOLE MANUSCRIPT.

Reviewer 3 Report

The authors presented an interesting visualization study of a low-power ns pulsed atmospheric pressure plasma jet. The observations are useful, and the qualitative analysis is sound. The article will generate interest in the plasma medicine and plasma-liquid interactions communities, in particular. The article will be suitable for publication after amendment.

Abstract: The authors should me more explicit and report key findings specific to each surface. 

Abstract: There is a missing "," in line 16. "... metal, dielectric..."

Line 66: Delete "adopted"

Line 95: Specify what intensity refers to. Emission intensity...

Line 133 and 149: corrisponding => corresponding. There are several other typos in the text (a spell-check software should be used).

Line 152 (and somewhere else in the text): "Relevantly" is not the correct word.

Line 194: "Dumped" => Dissipated

Line 206: ... electrical properties...

Line 228: "charging" => changing

Line 233: "... results visible in..." rephrase

The discussion (or conclusion) section should include a statement or two on the potential effects of flow turbulence generation on intended treatment, for instance. 

Round 2

Reviewer 2 Report

the authors have done a nice job in addressing my concerns. I now recommend publication of this paper in Plasma.